# Hypersensitivity Myocarditis after COVID-19 mRNA Vaccination

**DOI:** 10.3390/jcm11061660

**Published:** 2022-03-16

**Authors:** Andrea Frustaci, Romina Verardo, Nicola Galea, Carlo Lavalle, Giulia Bagnato, Rossella Scialla, Cristina Chimenti

**Affiliations:** 1Department of Clinical, Internal, Anesthesiologist and Cardiovascular Sciences, La Sapienza University, 00161 Rome, Italy; cristinachimenti@libero.it; 2Cellular and Molecular Cardiology Lab, IRCCS L. Spallanzani, 00149 Rome, Italy; romina.verardo@inmi.it (R.V.); giulia.bagnato@inmi.it (G.B.); rossella.scialla@inmi.it (R.S.); 3Department of Experimental Medicine, Sapienza University, 00161 Rome, Italy; nicola.galea@uniroma1.it; 4Department of DAI Cardio-Thoraco-Vascular and Organ Transplant Surgery, La Sapienza University, 00161 Rome, Italy; carlo.lavalle@uniroma1.it

**Keywords:** eosinophilic myocarditis, COVID-19 mRNA vaccination, myocarditis

## Abstract

Background: Myocarditis, even in a severe and lethal form, may occur after COVID-19 mRNA (BNT162b2) vaccination. However, its pathway, morphomolecular characterization and treatment are still unknown. Methods: Routine hematochemical screening, ECG, Holter monitoring, 2D echocardiogram cardiac magnetic resonance (CMR) and invasive cardiac studies (cardiac catheterization, selective coronary angiography, left ventriculography and left ventricular endomyocardial biopsy) are reported from three patients (39F-pt1, 78M-pt2, 52M-pt3) with severe compromise of conduction tissue (junctional rhythm and syncope, pt1) or cardiac function compromise (LVEF ≤ 35%, pt2 and pt3) after COVID-19 mRNA (BNT162b2). Results: Hematochemical data and coronary angiography were normal in the patients studied. Histology showed in all three patients extensive myocardial infiltration of degranulated eosinophils and elevation of serum cationic protein directly responsible for cardiomyocyte damage. These findings demonstrate myocarditis hypersensitivity to some component of the vaccine (spike protein?) acting as a hapten to some macromolecules of cardiomyocytes. Steroid administration (prednisone, 1 mg/kg die for 3 days, followed by 0.33 mg/kg for 4 weeks) was followed by complete recovery of cardiac contractility in pt2 and pt3. Conclusions: Eosinophilic myocarditis is a possible adverse reaction to the mRNA COVID-19 vaccine. Its pathway is mediated by release of cationic protein and responds to short courses of steroid administration.

## 1. Introduction

Reports of myocarditis after COVID-19 mRNA (BNT162b2) vaccination have raised concerns about the potential cardiovascular risk of this vaccine enhancing the skepticism of the no-vax population.

Indeed, although the incidence of acute inflammatory myocardial and/or pericardial disease has been documented in as low as 1–2 events in 100,000 vaccinated subjects [1,2], severe cases requiring hospitalization because of acute heart failure or progressing to death [3] have been described.

Morphomolecular characterization of this entity and its treatment, as well as the possibility to distinguish it from other (i.e., viral, autoimmune) forms of myocarditis, are not available so far.

We report herein the endomyocardial biopsy study of three patients with severe post-vax myocarditis, where a mechanism of hypersensitivity to some component of the vaccine is suggested, and an increase in serum eosinophilic cationic protein is recognized as responsible for tissue damage, as well as a potential biomarker of the disease.

## 2. Study Population

In the last 6 months, 3 patients (39F-pt1, 78M-pt2, 52M-pt3) were identified who, within a period of two weeks from the second dose of the BNT162b2 vaccine, manifested chest pain with troponin I elevation. A 39-year-old woman (pt1) also had two syncopal episodes because of junctional rhythm (Figure 1), and two men (pt2, pt3) showed severe compromise of myocardial contractility (left ventricular ejection fraction ≤ 35%). Before vaccination, all three subjects were in good health and denied previous major diseases but minor allergic reactions not requiring prolonged and specific medications.

## 3. Methods

All 3 patients underwent noninvasive clinical examinations (including routine hematochemical screening, resting ECG, Holter monitoring, 2D echocardiogram and cardiac magnetic resonance (CMR) and invasive cardiac studies (cardiac catheterization, selective coronary angiography, left ventriculography and left ventricular endomyocardial biopsy)).

An electrophysiological study (EPS) was performed in pt1 with an electroanatomic mapping system (Carto^®^ 3, Biosense Webster, Irvine, CA, USA). Via the transfemoral approach, a decapolar catheter was positioned in the coronary sinus, and a multipolar catheter (Pentaray; Biosense Webster Inc) was positioned in the right atrium (RA). EPS showed normal anterograde conduction parameter (HV 48 ms) with junctional rhythm without evidence of retrograde atrial activation. Pacing from high RA did not result in atrial capture, and the decapolar catheter did not record atrial electrograms from inside the coronary sinus. High-density mapping with the Pentaray catheter revealed the absence of endocavitary electrograms in almost the whole RA, with evidence of organized atrial activity in a small area in the anterolateral portion of the atrium (9.5 cmq), accounting for 4.9% of the total RA surface. Due to the absence of atrial activity, activation mapping was not performed. High-density substrate mapping disclosed a large low-voltage area (<0.05 mV), with a small region on the anterolateral aspect of RA with preserved voltage signals.

CMR, performed at baseline and one-month follow-up, was aimed at obtaining references on the dimensions of the heart chambers, wall thicknesses and right and left ventricular function. The CMR also provides possible alterations of tissue composition through the acquisition of T1 and T2 mapping and of late gadolinium enhancement (LGE). In particular, the increase in T2 signal is due to the myocardial edema detector, and the positivity for areas of LGE suggests the presence of inflammatory infiltration and/or myocardial fibrosis.

## 4. Endomyocardial Biopsy Study

For histological analysis, endomyocardial biopsy was fixed in 10% buffered formalin and embedded in paraffin. Section 5 microns thick were stained with hematoxylin/eosin and Masson’s trichrome.

The histological diagnosis of myocarditis was based on the evidence of leukocyte infiltrates (≥14 leukocytes/2 mm) associated with necrosis of the adjacent myocytes, according to the Dallas criteria [4] and confirmed by immunohistochemistry [5]. In particular, for the phenotypic characterization of inflammatory infiltrates, immunohistochemistry was performed for CD3, CD20, CD43, CD45RO and CD68 (all Dako, Carpinteria, CA, USA).

Immunohistochemical staining with eosinophil major basic protein (EMBP) antibody (Santacruz Biotechnology, Inc., Dallas, TX, USA) was performed to highlight the presence of eosinophils.

For histological characterization of inflammatory lesions being observed in post-COVID-19 myocarditis, endomyocardial biopsies with a diagnosis of myocarditis associated with COVID-19 infection were used for comparison.

## 5. Molecular Study

The molecular study was performed in the three patients with real-time (RT) PCR [6] for the SARS-CoV-2 virus and for the most common cardiotropic viruses (adenovirus, enterovirus, influenza A and B virus, Epstein–Barr virus, parvovirus B19, hepatitis C virus, cytomegalovirus, human herpes virus 6, herpes simplex type 1 and 2) for the possible identification of viral genomes.

## 6. Serologic Study

Patient serum was screened for the presence of human eosinophil cationic protein (ECP), detected by the human eosinophil cationic protein ELISA assay kit ECP ELISA (MBS700481, MyBioSource San Diego, CA, USA), according to the manufacturer’s instructions. The standards, sample and healthy control were loaded in duplicate.

## 7. Statistical Analysis

Statistical analysis was performed by the GraphPad Prism package, version 5.02 (GraphPad Software Inc., San Diego, CA, USA). Comparison between groups was performed with the Mann–Whitney test, and a value of *p* < 0.05 was considered statistically significant.

## 8. Results

Hematochemical screening beyond elevation of troponin I (3.5 ± 0.2 mcg/L ± n.v 0.1 ± 0.14 mcg/L) was unremarkable. In particular, normal was the leukocyte count and the peripheral amount of eosinophils.

Cardiac catheterization showed elevation of LV end-diastolic pressure (>12 Hg mm), while coronary angiography was normal in all cases.

EPS obtained in pt1 with junctional rhythm showed normal anterograde conduction parameter (HV 48 ms), with junctional rhythm without evidence of retrograde atrial activation. Pacing from high RA did not result in atrial capture, and the decapolar catheter did not record atrial electrograms from inside the coronary sinus. High-density mapping with the Pentaray catheter revealed the absence of endocavitary electrograms in almost the whole RA, with evidence of organized atrial activity in a small area in the anterolateral portion of the atrium (9.5 cmq), accounting for 4.9% of the total RA surface. Due to the absence of atrial activity, activation mapping was not performed. High-density substrate mapping disclosed a large low-voltage area (<0.05 mV), with a small region on the anterolateral aspect of RA with preserved voltage signals (Figure 1).

CMR showed in our three patients a diffuse increase in both T2 and T1 myocardial value, as a result of diffuse inflammatory involvement (Figure 2 and Figure 3). LGE was present as well and patchily distributed in the subepicardial myocardium. Severe compromise of LV contractility of 35 and 28% was registered in pt2 (Figure 3) and pt3, respectively.

## 9. Histology

Endomyocardial biopsy findings were characterized in all three cases by the presence of inflammatory infiltrates, mainly represented by degranulated eosinophils and to a very less amount of lymphocytes (Figure 2) focally associated with necrosis of the adjacent myocytes. In a sample from a pt1 manifesting junctional rhythm and no atrial activity at electrophysiological study, peripheral sections of conduction tissue (Purkinje fibers) were included, being infiltrated and damaged by eosinophils (Figure 2).

## 10. Serologic Study

Patient serum showed an increase in serum cation protein levels in all patients (23.4 ± 17 U/mL) versus controls (6.1 ± 1 U/mL), *p* = 0.079.

## 11. Molecular Biology

RT-PCR for viral genomes was negative in all patients.

## 12. Treatment and Follow-Up

The three patients received prednisone 1 mg/kg die for 3 days tapered to 0.33 mg/kg for 4 weeks. At the end, a new noninvasive cardiac evaluation, including ECG, Holter and cardiac magnetic resonance, was undertaken. Pt1 remained in junctional rhythm but without abnormal pauses and requiring pacemaker implantation. Pt2 and pt3 manifested a complete recovery of cardiac function, with LVEF rising to 49.5 % (from 35%) and 56% (from 28%), respectively.

## 13. Discussion

Myocarditis is one of the major recognized adverse reactions to the COVID-19 mRNA (BNT162b2) vaccine. Although large reports confine its incidence to 1–2 cases in 100,000 administrations, severe and even lethal cases of acute myocardial inflammation have been described [3]. Nevertheless, morphomolecular characteristics of these inflammatory lesions, their relationship with myocarditis associated with COVID-19 infection and their possible treatment are not known so far.

The present report describes three cases of acute severe myocarditis occurring within 2 weeks after a second dose of the COVID-19 mRNA (BNT162b2) vaccine. Major clinical manifestations consisted of damage of conduction tissue with junctional rhythm and atrial standstill or remarkable compromise of left ventricular function (EF ≤ 35%) that made necessary an invasive cardiac study. The histological findings obtained by left ventricular endomyocardial biopsy were characterized in all instances by presence in the myocardium of prevalent eosinophilic infiltrates (Figure 2G), associated with degranulation of crystalloids and elevation in the circulatory blood of cationic protein. This last protein is known in patients with eosinophilic endomyocardial disease to induce myocardial and coronary vessel damage (Churg–Strauss syndrome), as well as endocarditis with thrombus formation (Loeffler disease) because of parallel activation of factor X of coagulation.

Post-vax inflammatory lesions observed in our patients demonstrate myocarditis hypersensitivity and the formation of new antigens from macromolecules of cardiomyocytes and some component (spike protein?) of the BNT162b2 vaccine.

Excellent response to moderate doses of steroids with recovery, in 4 weeks, in two cases with severely compromised cardiac function supports this interpretation. Furthermore, presence of eosinophils in myocardial infiltrates has already been described in previous reports of post-COVID19 vax myocarditis undergoing endomyocardial biopsy or post-mortem study [3]. However, their biological significance, as well as their potential pathogenetic role in myocardial damage, has not been investigated. Finally, prevalent eosinophilic myocarditis has already been described for other forms of vaccine after smallpox vaccination [7].

Interestingly, all three patients described in our report were affected by allergic disorders that would indicate some predisposition to allergic reactions to new antigens. Although the type of antigen is so far unknown, a haptenic link between spike protein and macromolecules of cardiomyocytes membrane can be hypothesized.

Finally, comparison with biopsy and post-mortem myocarditis associated with COVID-19 infection indicates (Figure 3H) a substantial difference with our description, showing CD45Ro-positive lymphocytic infiltrates becoming responsible for virus-negative immune-mediated damage of cardiomyocytes, intramural vessels, conduction tissue and even subepicardial ganglia [8].

If our preliminary results are confirmed by larger studies, elevation of serum cationic protein could be taken as a biomarker of hypersensitivity myocarditis, suggesting the introduction of steroid treatment in patients with clinical and magnetic resonance evidence of perimyocarditis developed after COVID-19 mRNA vaccination.

In conclusion, eosinophilic myocarditis is a possible adverse reaction to the mRNA COVID-19 vaccine. Its pathway is probably mediated by release of cationic protein and responds to short courses of steroid administration.

## Figures and Tables

**Figure 1 jcm-11-01660-f001:**
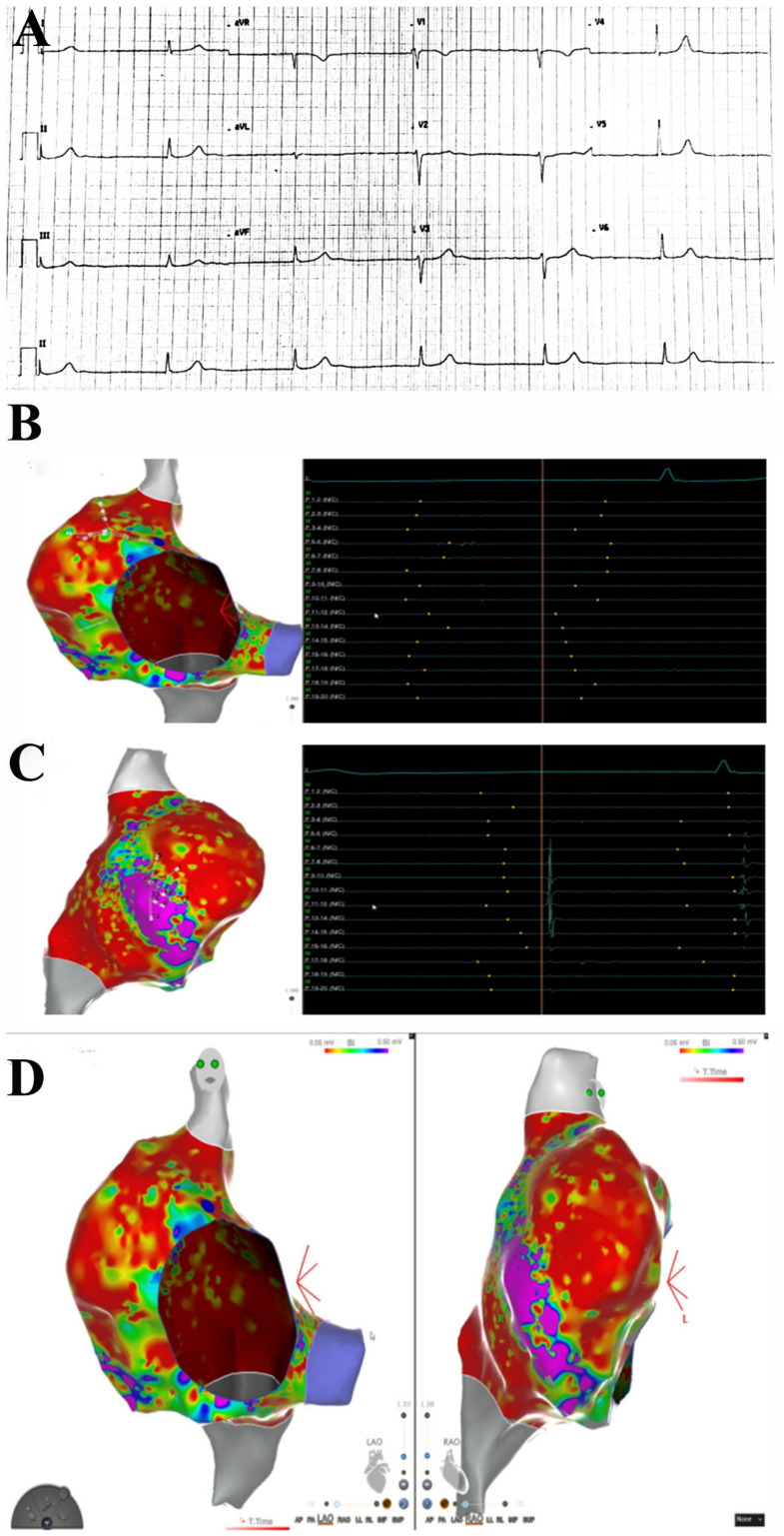
Right atrial substrate mapping and endocavitary electrogram recordings. ECG shows no sinus rhythm, no atrial activity at EPS. Junctional rhythm at 40 bts/min (**A**). Endocavitary recordings show the absence of atrial activation during atrial mapping (**B**). Some atrial signals are recorded on the anterolateral wall of right atrium (**C**). Left and right anterior oblique projections of right atrium endocardial substrate mapping shows a large scarred area with low-voltage signals (<0.05 mV, in red), with a small portion of myocardial wall with preserved voltage signals (>0.5 mV, in purple) on the anterolateral aspect of right atrium (**D**).

**Figure 2 jcm-11-01660-f002:**
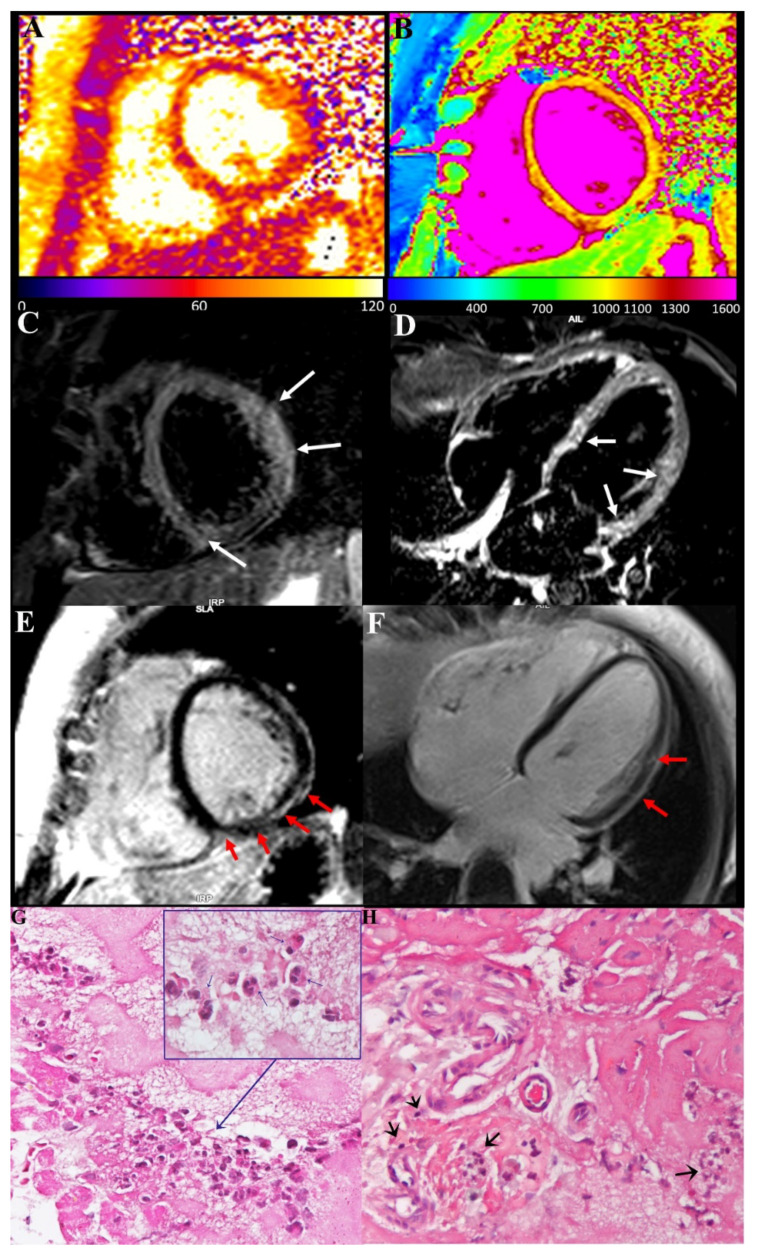
Eosinophilic myocarditis with conduction tissue involvement after COVID-19 mRNA vaccination. (**A**,**B**): Myocardial T2 (**A**) and T1 (**B**) maps, acquired on mid-ventricular short-axis view, showed a diffuse increase in both T2 and T1 myocardial value as a result of a diffuse inflammatory involvement. (**C**,**D**): Presence of myocardial edema was also confirmed on STIR T2-weighted images acquired on short-axis (**C**) and horizontal long-axis (**D**) views as multiple areas of hyperintense signal within the basal septum, the inferior interventricular junction and mid-basal anterolateral wall (arrows). (**E**,**F**): Late-enhancement images acquired on the same short-axis (**E**) and horizontal long-axis (**F**) planes reveal tiny areas of mild enhancement with “patchy” distribution and predominant involvement of subepicardial layer in the same location of the above-mentioned myocardial edema (red arrows). (**G**,**H**): Histology shows degranulation of crystalloids of eosinophilic granulocytes (see insert and blue arrow) responsible for the release of the cationic protein and ultimately for myocardial damage and a strongly infiltration of eosinophils in the conduction tissue (**H**) (black arrow). (400× magnification).

**Figure 3 jcm-11-01660-f003:**
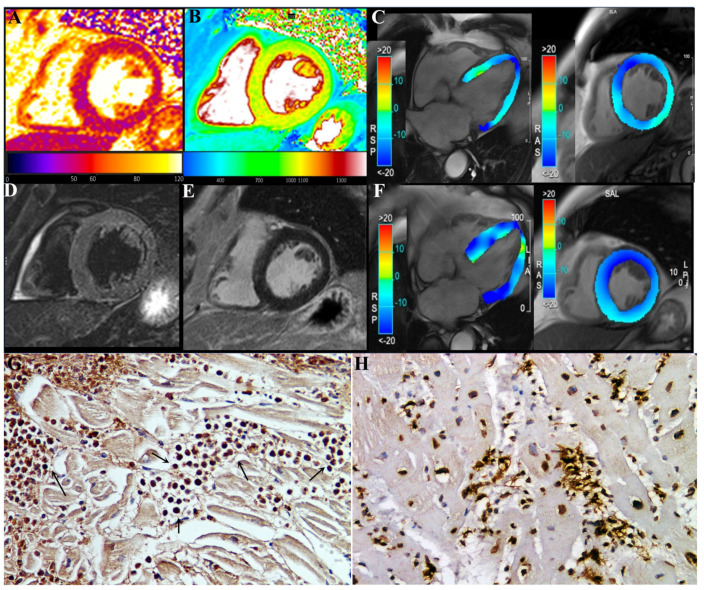
A 78-year-old male (pt2) with eosinophilic myocarditis myocarditis after second dose of COVID-19 mRNA (BNT162b2) vaccine. At first CMR exam (**A**–**E**), performed 4 weeks after symptoms onset, T2 (**A**) and native T1 (**B**) maps revealed an increase in global T2 (55 ms ± 4 ms, normal range < 49.9 ms) and T1 (1060 ± 15 ms, normal range < 1027 ms) myocardial values, without any noticeable focal areas of signal changes on STIR T2-weighted (**D**) and late-gadolinium-enhanced (**E**) images, reflecting mild diffuse edematous condition. CineMR images, acquired on four-chamber (left) and mid-ventricular short-axis (right) views, analyzed by tissue tracking analysis show moderate impairment of both longitudinal and circumferential systolic left ventricular function (GLS: −12.3%, GCS: −13.6%, EF: 35%) at admission (**C**), which improved at two-month follow-up after therapy ((**F**), GLS: −18.1%, GCS: −17.8%, EF: 49.5%). GCS: global circumferential strain; GLS: global longitudinal strain; EF: ejection fraction. (**G**) shows in post-vax myocarditis massive eosinophilic myocardial infiltration with necrosis of the adjacent myocytes (IHC for eosinophil major basic protein (EMBP) antibody). (200× magnification). (**H**): Comparison with acute myocarditis associated to COVID19 infection showing intense lymphocytic inflammation (IHC for CD45Ro) (200× magnification).

## Data Availability

The datasets used and analyzed during the current study are available from the corresponding author upon reasonable request.

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
