# Peer review of "Hypersensitivity Myocarditis after COVID-19 mRNA Vaccination"

_jcm, 2022, doi:10.3390/jcm11061660_

Round 1

Reviewer 1 Report

Congratulations, very interesting study that opens new perspectives in the diagnosis and treatment of myocarditis post covid vaccine. I point out some typos and small suggestions: line 66: (RA) instead of (RS) line 87: ...of myocarditis was based... line 89: ...and confirmed... line 105: ...detected by an ELISA... line 110-111: "Comparison between groups was performed with Mann Whitney non parametric test was used." Is not clear. line 114: ...elevation... line 155: ...1-2... line 182-183: It would be interesting to know the type of allergic reaction and the potential allergen
line 192: Perimyocarditis would be more correct
line-193-195: In the conclusions, given the number of patients, however, in a study conducted very well I would leave room for a probability rather than a certainty, ie ...Its pathway is probably mediated by release...

Author Response

Reply: We Thanks to the reviewer for his/her comments. All suggestions have been introduced in the text (in bold).

Reviewer 2 Report

Andrea Frustaci and colleagues showed that morpho-molecular characterization of myocarditis after Covid-19 mRNA vaccination. In particular, the histological and serum results indicating hypersensitive myocarditis are impressive.

Overall, the findings are interesting, and the data appear to be of good quality, but the authors need to present their findings in more detail.

Major and minor comments are as follows:

Major comments

  1. Have the cross-reactivities between cardiomyocytes and spike protein of Corona virus or vaccine proteins been reported previously?
  2. In patient 1, why did inflammation affect right atrium specifically? What's the mechanism?
  3.  All three patients have past history of allergic diseases. Is the increase in serum cationic protein significant? Did not that indicate a rise at baseline?

Author Response

Reply: We thank the reviewer for his/ her considerations.

Major comments

Have the cross-reactivities between cardiomyocytes and spike protein of Corona virus or vaccine proteins been reported previously?

Reply Not yet in our knowledge.

In patient 1, why did inflammation affect right atrium specifically? What's the mechanism?

Reply: We are actually unable to answer this question. Nevertheless, Severe bradycardia as well as sino-atrial node dysfunction has been frequently noted in pts with post-vax Covid 19 myocarditis.

 All three patients have past history of allergic diseases. Is the increase in serum cationic protein significant? Did not that indicate a rise at baseline?

Reply: We did not have previous data on serum levels of cationic protein in our patients. The remarkable and rapid response to steroid treatment suggest an its likely pathogenetic role.

Round 2

Reviewer 2 Report

Andrea Frustaci and colleagues showed that morpho-molecular characterization of myocarditis after Covid-19 mRNA vaccination.

In particular, the histological and serum results indicating hypersensitive myocarditis are impressive.

Overall, the findings are interesting, and the data appear to be of good quality. Authors have responded to our comments appropriately.